# Preparation of Low-Temperature Solution-Processed High-κ Gate Dielectrics Using Organic–Inorganic TiO_2_ Hybrid Nanoparticles

**DOI:** 10.3390/nano14060488

**Published:** 2024-03-08

**Authors:** Hong Nhung Le, Rixuan Wang, Benliang Hou, Sehyun Kim, Juyoung Kim

**Affiliations:** 1Nanocomposite Structure Polymer Laboratory, Department of Advanced Materials Engineering, Kangwon National University, Samcheok 25913, Republic of Korea; hongnhungle94@gmail.com; 2School of Chemical Engineering, Yeungnam University, Gyeongsan 38541, Republic of Korea; nl910213@gmail.com; 3Division of Chemical Engineering, Konkuk University, Seoul 05029, Republic of Korea; houbenliang0330@gmail.com

**Keywords:** organic–inorganic, hybrid, colloidal, stable, titania, gate dielectric, OTFT, coating, amphiphilic polymer

## Abstract

Organic–inorganic hybrid dielectric nanomaterials are vital for OTFT applications due to their unique combination of organic dielectric and inorganic properties. Despite the challenges in preparing stable titania (TiO_2_) nanoparticles, we successfully synthesized colloidally stable organic–inorganic (O-I) TiO_2_ hybrid nanoparticles using an amphiphilic polymer as a stabilizer through a low-temperature sol–gel process. The resulting O-I TiO_2_ hybrid sols exhibited long-term stability and formed a high-quality dielectric layer with a high dielectric constant (κ) and minimal leakage current density. We also addressed the effect of the ethylene oxide chain within the hydrophilic segment of the amphiphilic polymer on the dielectric properties of the coating film derived from O-I TiO_2_ hybrid sols. Using the O-I TiO_2_ hybrid dielectric layer with excellent insulating properties enhanced the electrical performance of the gate dielectrics, including superior field-effect mobility and stable operation in OTFT devices. We believe that this study provides a reliable method for the preparation of O-I hybrid TiO_2_ dielectric materials designed to enhance the operational stability and electrical performance of OTFTs.

## 1. Introduction

Recently, printed electronics have provided significant advancements in materials and devices [1,2,3]. These advancements have opened up diverse applications in fields such as flexible displays [4], lightweight wearable devices [5], and low-cost disposable circuits [6] due to their low-temperature- and solution-processing, allowing for cost-effective large-area production and compatibility with flexible substrates. In this technological context, solution-processable high-performance electronic materials have emerged as a focal point in materials science research. Specifically, these materials have been extensively studied for their potential application in thin-film transistors (TFTs), versatile building blocks for various devices [7,8]. Organic TFTs (OTFTs), which utilize solution-processable organic semiconductors and dielectric materials, offer numerous competitive benefits. These include cost-effective and high-throughput manufacturing, exceptional inherent flexibility, and compatibility with a wide range of substrates such as plastic films and papers [2,9,10,11]. This compatibility facilitates the continuous integration of multifunctional materials in hybrid structures. Particularly, high-dielectric-constant (high-κ) materials are required in manufacturing dielectric insulators as a major component in microelectronic devices such as central processing units (CPUs), dynamic random-access memory (DRAM), and flash memory. The primary function of the dielectric material is to regulate the conductance of the semiconducting channel by storing charge carriers. Additionally, the dielectric material must possess electrical insulation properties to minimize leakage current and static dissipation. Hence, an ideal high- κ material must satisfy the following criteria: (1) a sufficiently high dielectric constant for sustainable scaling; (2) a sufficiently large band gap (E_g_ > 5 eV) and appropriate band offsets with a substrate of over 1 eV; (3) exceptional thermodynamic and kinetic stability with the substrate; (4) compatibility with current IC processes; and (5) a minimal density of electrically active defects in the bulk.

Conventionally, solution-processed polymer dielectrics have been intensively studied for OTFTs because of their ability to be processed at low temperatures and excellent mechanical flexibility [12,13]. However, these polymer dielectrics often have relatively low dielectric constants, limiting their suitability for certain microelectronic devices. To address this, solution-processed SiO_2_ dielectric materials have emerged as potential candidates due to their defect-free thin film growth, excellent optical transparency, and good mechanical properties. As the integration level of microelectronic devices increases, the thickness of SiO_2_ needs to decrease to maintain device performance (thinner than ~1 nm), leading to challenges such as leakage current due to quantum tunneling at very thin SiO_2_ layers. The challenge has been addressed by replacing SiO_2_ with insulators that have high dielectric constants. By using high-κ dielectrics, it is possible to increase the dielectric thickness while maintaining the same capacitance, effectively reducing the leakage current. Currently, the favored high-κ dielectrics are HfO_2_ [14], TiO_2_, ZrO_2_, and Al_2_O_3_ [15,16,17]. Nevertheless, the mechanical durability of metal oxide (MO) dielectrics is somewhat lacking for use in flexible electronics due to their inherently brittle nature, which hinders the full use of MO dielectrics in emerging flexible electronics. Additionally, the fabrication of solution-processed MOx dielectric materials often requires high-temperature annealing (>400 °C) and an extended processing time, which is unsuitable for industry-scale commercialization. Therefore, fabricating materials by incorporating inorganic components and organic components can be a pathway to designing dielectric materials with the desired properties.

In the early days, high-κ MO nanoparticles (NPs) dispersed uniformly in a polymer matrix, called high-κ dielectric nanocomposite materials, were prepared to solve these problems [18]. However, another problem is that the quality of nanocomposites as gate dielectric materials largely depends upon the uniformity with which the NPs are dispersed in the polymer matrix. Unevenly distributed NPs can increase the surface roughness of a gate dielectric, reducing the mobility due to surface scattering. Moreover, the incorporation of MOx nanoparticles into the polymer matrix could lead to the formation of pinholes in the film and/or film cracking due to undesirable aggregation effects, preventing the dielectric thickness scaling. Hence, organic–inorganic hybrid dielectric materials have been prepared via the sol–gel process to overcome these limitations, providing ample potential for high-performance flexible electronics [19,20,21]. For example, Gheona et al. [22] detailed the synthesis of a sol-gel-processed organic-inorganic hybrid incorporating phosphorus and zirconium. The resultant material, a solid jelly-like substance, exhibits a wide spectrum of properties, ranging from soft and pliable to hard, thus offering significant potential for various applications. This can be explained by the combination of the organic dielectrics providing outstanding mechanical flexibility and the inorganic dielectrics contributing to good insulating and dielectric properties. The main advantage in the fabrication of organic–inorganic hybrid dielectrics is the proper cross-linking process of organic materials and MOx materials at the molecular-scale level via physical interaction (hydrogen bonds, van der Waals, and ionic interaction) or chemical interaction (covalent bond). Coating films prepared with organic–inorganic hybrid dielectric materials could not only solve the low dielectric problem but also show high flexibility, durability, and adhesive properties, which could be used for developing flexible dielectrics, one of the most important components in manufacturing flexible and rollable electronic devices [23]. 

Currently, titania (TiO_2_) is receiving much attention as an excellent candidate for preparing high-κ dielectric materials due to its high dielectric constant, high refractive index, and good mechanical properties. However, achieving colloidally stable TiO_2_ nanoparticles for the preparation of solution-processed gate dielectrics remains a significant challenge due to the high reactivity of titanium precursors, leading to easy aggregation during the hydrolysis–condensation reaction. Consequently, there are only a few studies on the preparation of colloidally stable O-I TiO_2_ nanosols [24,25]. Even when these solutions have been prepared, the dielectric constants of their coating films have not been sufficiently high (lower than 10) to produce excellent OTFT devices [26,27,28]. 

In this study, we present a simple and low-temperature method for the preparation of colloidally stable O-I TiO_2_ nanosols. In the presence of an amphiphilic urethane precursor (AUP) as a stabilization agent, we not only achieved the colloidal stability of TiO_2_ nanoparticles but also facilitated the formation of a high-quality thin film derived from them, which was used as a gate dielectric. The AUP comprises a hydrophilic chain (polyethylene oxide—PEO) and a hydrophobic chain (polypropylene oxide—PPO) on the same backbone, with the hydrophilic chain playing a crucial role in stabilizing the silica nanoparticles. The length of this hydrophilic chain strongly impacts the dielectric properties of the coating film derived from these nanoparticles due to its influence on the size distribution, interfacial interaction, and water absorption characteristics [29,30]. To clarify these effects, we used three types of AUP with varying lengths of the hydrophilic chain while keeping the hydrophobic chain constant to prepare different types of O-I TiO_2_ hybrid nanoparticles. We also addressed common issues of nanocomposite gate dielectrics. After assessing the insulating properties of the organic–inorganic hybrid TiO_2_ dielectric layers, we conducted performance tests on OTFT devices incorporating the best dielectric properties. We expect the promising potential of the prepared O-I TiO_2_ hybrid nanoparticles to enhance the performance of OTFTs.

## 2. Materials and Methods

### 2.1. Materials

For the synthesis of the amphiphilic urethane precursors (AUPs), 2,4-toluene diisocyanate (TDI; Mw = 174.2 g/mol, Sigma-Aldrich Co., St. Louis, MO, USA), glycerol propoxylate (GP, Mw = 1000 g/mol, KPX Chemical Co., Seoul, Republic of Korea), 2-hydroxyethyl methacrylate (2-HEMA, Mw = 130.143 g/mol, Sigma-Aldrich Co., St. Louis, MO, USA), poly(ethylene glycol) methyl ether (mPEG, Mw = 550 g/mol, Mw = 750 g/mol, Mw = 2000 g/mol, Sigma Aldrich, USA), and acetone (Ac, Sigma Aldrich, USA) were used as received. For the preparation of the organic–inorganic hybrid solutions, titanium(IV) isopropoxide (TTiP, Mw = 284.22, Sigma Aldrich, USA) was used as the inorganic precursor. Methyl ethyl ketone (MEK, Sigma Aldrich, USA) and ethanol (EtOH, Sigma Aldrich, USA) were used as solvents without further purification. Azobisisobutyronitrile (AIBN, Sigma Aldrich, USA) was used as an initiator in the polymerization of the AUPs. Finally, 0.1 M HCl (aq) was used to carry out the hydrolysis reaction. For the preparation of TiO_2_/poly(AUP) nanocomposites, commercial anatase TiO_2_ nanopowders (5 nm, SkySpring Nanomaterials, Inc., Houston, TX, USA) and alkaline hydrogen peroxide (H_2_O_2_, 30 wt%, Daejung Chemical Co., Siheung, Republic of Korea) were used without any further purification. Moreover, 1.6 M HNO_3_ and deionized water (DI) were used in the washing step. Finally, for device preparation, acetone (Ac, Sigma Aldrich, St. Louis, MO, USA), isopropanol (99.9%, Duksan pure Chemicals Co., Ltd., Ansan, Republic of Korea), and 2-methoxyalcohol (99.8%, Sigma Aldrich, St. Louis, MO, USA) were used as received.

### 2.2. Synthesis of AUPs

The AUPs were synthesized through a three-stage reaction process. A schematic presentation of the AUP’s chemical structure is shown in Figure 1. The steps were performed using the same synthesis procedure of urethane amphiphilic nonionomer (UAN) as described in detail in our previous studies [31,32]. The Arabic number in the AUP name represents the molecular weights of GP and mPEG used in the synthesis process. For example, AUP 1000-m550 is a precursor synthesized using GP with a molecular weight of Mw = 1000 g/mol and mPEG with a molecular weight of Mw = 550 g/mol on the same backbone. Moreover, the letter “m” indicates that mPEG was used in the AUP synthesis.

### 2.3. Preparation of O-I TiO_2_ Hybrid Nanoparticles

At first, 6 g of AUP was mixed with 30 g of MEK and polymerized in the presence of 0.07 g of AIBN under a nitrogen atmosphere at 60 °C for 12 h. The resulting product after polymerization was named poly(AUP). Then, 8 g of poly(AUP) solution was mixed with 7 g of TTiP in 30 g of ethanol used as a solvent. In this study, we prepared 3 types of poly(AUP) solutions, namely, poly(AUP) 1000-m550, poly(AUP) 1000-m750, and poly(AUP) 1000-m2000, corresponding to AUP 1000-m550, AUP 1000-m750, and AUP 1000-m2000. Following that, an aqueous solution of 0.1 M HCl_aq_ was added to the mixture (where the molar ratio of H_2_O/Ti was 3) and stirred at 60 °C for 36 h to complete the hydrolysis–condensation reaction. The pH of the solution was adjusted to 2.78 at 25 °C condition with 37 wt% HCl to prevent the formation of precipitates. The prepared O-I TiO_2_ hybrid sols were referred to as UTi7-m2000, UTi7-m750, and UTi7-m550, where the number 7 before the dash indicates the weight ratio between the titanium precursor and AUP, and the number after the dash represents the molecular weight of the mPEG chain used in the synthesis of the AUP.

### 2.4. Preparation of TiO_2_/poly(AUP) Nanocomposites

First, commercial anatase TiO_2_ nanopowder was dispersed in an alkaline hydrogen peroxide solution. The weight percentage of the mixture accounted for 30%. The mixture was stirred constantly for 36 h at 50 °C. Subsequently, a white sodium titanate precipitate was formed and washed using a 1.6 M HNO_3_ solution. The resulting precipitate was thoroughly washed several times with deionized water (DI) to remove any remaining ionic impurities and subsequently dried in a vacuum freeze dryer for 24 h at −110 °C. The surface-treated TiO_2_ nanopowders were placed in 70 mL glass vials and stored at room temperature for further use. 

Next, 1 g of poly(AUP) 1000-m550 (prepared by polymerization of 6 g of AUP 1000-m550 in 30 g of MEK with 0.07 g of AIBN as initiator) was dissolved in ethanol. Next, different percentages of the surface-treated TiO_2_ nanopowders (5, 10, and 15 wt% on dried polymer) were introduced into the poly(AUP) solution, and the mixtures underwent sonication for one day. The resulting solutions were named TiO_2_/poly(AUP)—5 wt%, TiO_2_/poly(AUP)—10 wt%, and TiO_2_/poly(AUP)—15 wt%, corresponding to the weight percentage of TiO_2_ within the solution.

### 2.5. Sample Preparation (Dielectric Deposition and Device Fabrication)

#### 2.5.1. Fabrication of Metal–Insulator–Metal (MIM) Device

N-type (100) silicon wafers (resistivity < 0.005 Ω·cm, Namkang Hi-Tech Co., Ltd., Seongnam, Republic of Korea) were used as substrates. The silicon wafers were first cleaned in boiling acetone for 30 min. Subsequently, an ultrasonic cleaner was used for sequential cleaning with acetone and isopropanol for 30 min each. After cleaning, the wafers were dried with industrial nitrogen and further cleaned by exposure to UV/ozone for 30 min. Metal shadow masks were utilized to evaporate a 30 nm thick Al gate electrode on each substrate (deposition rate = 15 Å·s^−1^; vacuum pressure = 10^−6^ Torr; substrate temperature = 25 °C). The prepared materials were dissolved in 2-methoxyalcohol (2-Me) and vigorously stirred for 2 h in ambient air. The prepared solutions were spin-coated onto substrates at 4000 rpm for 60 s and then thermally annealed at 120 °C for 30 min. Finally, an Al gate electrode was applied again, following the cross structure to complete the MIM device.

#### 2.5.2. Fabrication of OTFT Device

For the gate-patterned samples, the UTi7-m550 dielectric layer was deposited using a spin-coating process (5000 rpm, 30 s). Subsequently, the coated samples underwent annealing at 110 °C for 30 min in ambient air. The semiconductor (2,9-di-decyl-dinaphtho [2,3-b:2′,3′-f]-thieno-[3,2-b]-thiophene (C10-DNTT, Lumtec)) was deposited onto the gate dielectric layer by using organic molecular beam deposition (deposition rate = 0.1 Å·s^−1^; vacuum pressure = 10^−6^ Torr; substrate temperature = 25 °C) through a metal shadow mask. Lastly, the fabrication of the bottom-gate top-contact OTFT was finalized by thermally evaporating a 100 nm thick Au electrode layer onto the semiconductor layer (deposition rate = 2 Å·s^−1^; vacuum pressure = 10^−6^ Torr; substrate temperature = 25 °C). The channel length and width were 150 and 1500 μm, respectively.

### 2.6. Characterization

A Nicolet iS5 FT-IR spectrometer (Thermo Fisher Scientific, Waltham, MA, USA) was used to analyze the synthesized AUPs within the range of 4000–400 cm^−1^. Gel permeation chromatography (GPC, EcoSEC HLC-8320 GPC, Tosoh, Tokyo, Japan) was utilized to measure the molecular weights of the AUPs. DMF was used as the eluent for calibration and dissolving the amphiphilic polymer. The concentration of the sample used for measurement was 3 mg/mL, and 10 μL was injected at a flow rate of 0.35 mL/min at 40 °C. A dynamic light scattering analyzer (DLS, Zetasier Nano ZS, Malvern Instruments, Malvern, UK) was used to measure the particle size of the O-I TiO_2_ nanoparticles and the amphiphilic polymers dispersed in organic solvents. The thermal properties and inorganic contents of the UTi solutions were measured using thermogravimetric analysis (TGA, SDT Q600 V20.5 Build 15 Universal V4.4A, TA Instruments, New Castle, DE, USA). The analysis was carried out under a nitrogen (N_2_) atmosphere (flow rate of 20 Cc per minute) at a heating rate of 10 °C min^−1^ from room temperature to 800 °C. The observed mass loss was attributed to complete organic component decomposition, with the residual incombustible remains considered to represent an inorganic component (TiO_2_). An X-ray photoelectron spectrometer (XPS, K-Alpha, Thermo Fisher Scientific, Waltham, MA, USA) with monochromatic Al-KCl was used to identify the chemical constituents. The surface structure of the coating films was examined using a field emission scanning electron microscope (FE-SEM, JEOL JSM-6701F/X-MAX, Peabody, MA, USA). The roughness of the coating’s surface was measured using atomic force microscopy (AFM, multimode 8, Bruker, Karlsruhe, Germany). The thickness of the thin layers was measured through a cross-sectional scanning electron microscope (SEM, Hitachi SU8, Tokyo, Japan). Dielectric constant characteristics (capacitance values) were determined using an Agilent 4284 precision LCR meter (Agilent Technologies, Santa Clara, CA, USA.), and electrical properties were measured using a Keithley 4200 SCS (Keithley Instruments, Inc., Cleveland, OH, USA)unit in ambient air.

## 3. Results and Discussion

### 3.1. Characterization of AUPs

Figure 2 shows the structure analysis of three AUPs using Fourier transform infrared spectroscopy (FT-IR). It can be seen that the NCO peak at 2270 cm^−1^ was absent from the spectra of all of the three precursors, which indicates a 100% reaction of the NCO group of TDI with the OH groups of GP, 2-HEMA, and mPEG. The presence of urethane peaks in the range 1720–1730 cm^−1^, an acrylate peak at 1677 cm^−1^, and the bands around 840 cm^−1^ (C-O, C-C stretching, and CH_2_ rocking) demonstrate that the AUPs had urethane bonds, a PEO chain, and an acrylate group within them. Table 1 provides the molecular weights of the synthesized AUPs measured, along with the polydispersity index (PDI) obtained using GPC and the size of the particles of the AUPs dispersed in ethanol measured with DLS. The average molecular weights of the AUPs were in the range of 8996–38,765 g/mol, with a PDI of 1.32–2.1, indicating a broad molar mass distribution. Since the synthesized AUPs contain hydrophobic and hydrophilic segments in their structures, they can be easily dispersed in various solvents. Depending on the type of solvent, the size of the AUP nanoparticles dispersed in it may vary due to differences in solubility between the polyethylene oxide-based segment and the polypropylene-based segment, leading to microphase separation between these segments and resulting in the formation of nanoparticles dispersed in the solvent [33]. According to the DLS data, the particle size of the AUPs varied between 4.7 nm and 33.6 nm, proportional to the length of the hydrophilic chain. Based on this trend, it can be predicted that the size of O-I TiO_2_ hybrid nanoparticles will also be proportional to the length of the hydrophilic chain of the used amphiphilic polymers.

### 3.2. Preparation of Colloidally Stable O-I TiO_2_ Hybrid Sols

Preparation of nanoscale colloidal O-I TiO_2_ hybrid nanoparticles can be efficiently achieved by carrying out the hydrolysis and condensation of titanium alkoxides in aqueous environments. When water is present, alkoxides undergo hydrolysis and then polymerization to create a three-dimensional oxide network. Figure 3 shows the formation of O-I hybrid TiO_2_ nanoparticles under the hydrolysis–condensation process. At first, the titanol group (Ti-OH) was produced by the hydrolysis reaction of the TTiP monomers under acidic conditions, and its content was dependent on the r value (the molar ratio of Ti to H_2_O). Then, Ti-OH was converted into Ti-O-Ti crosslinks through a sequence involving the dealcoholation reaction, followed by the dehydration reaction of the formed Ti-OH groups. However, due to the high reactivity of the titanium precursor, the hydrolysis reaction in the presence of excess water is rapid and completes within seconds, causing the aggregation of TiO_2_ nanoparticles. Therefore, it is necessary to control the pH of the solution by using a stabilization agent. The pH of the solution played an important role in the stabilization of the titanium precursor. At pH 2–3 (near the point of zero charge of TiO_2_), the solution became strongly acidic, resulting in the slowing of the hydrolysis reaction of water with the precursor due to the highly acidic water of the suppressed OH groups. The slow addition allowed the partly hydrolyzed precursor molecules to disperse during stirring and avoid condensation to form precipitates [34]. Consequently, poly(AUP550), formed via the polymerization process of AUP, was added to the solution. The OH groups on the surface of the TiO_2_ nanoparticles could interact with the moieties’ oxide of the hydrophilic chain during the hydrolysis–condensation reaction through a hydrogen bond, preventing the aggregation of TiO_2_ nanoparticles.

The presence of the AUP enables the resulting TiO_2_ nanoparticles, after the hydrolysis–condensation reaction, to disperse in an organic solvent. Confirmation of the dispersion is verified through size measurements using dynamic light scattering analysis, as summarized in Table 2. It can be seen that the particle size increases with the molecular weight of mPEG: UTi7-m550 has a particle size of 4.354 nm, UTi7-m750 has a slightly larger size of 4.359 nm, and UTi7-m2000 exhibits the largest particle size at 9.703 nm. This trend is attributed to the impact of the mPEG chain length on the hydrophilic properties of the amphiphilic polymer. A longer mPEG chain, as seen in UTi7-m2000, provides a more extensive hydrophilic component. During the hydrolysis–condensation reaction, the hydrophilic chains on the nanoparticles’ surfaces may interact differently, affecting the overall structure and size of the particles. Longer mPEG chains could potentially lead to larger particles due to increased steric hindrance and altered intermolecular interactions during the formation process. Additionally, the effect of mPEG chains on the stability of nanoparticles in the long term is considered a crucial characteristic for achieving a reliable TFT operation and hence, commercialization of the TFT. As shown in Table 2, there was minimal change in the size of the TiO_2_ O-I hybrid nanoparticles even after six months of preparation and storage at ambient temperature, highlighting their potential for long-term stabilization. UTi7-m550 and UTi7-m750 showed slight changes in size (0.02–0.09 nm), whereas UTi7-m2000, which was stabilized by a polymer with the longest hydrophilic chain, exhibited a significant increase in size (from 9.70 nm after preparation to 10.3 nm after 6 months of storage at ambient temperature). This could be explained by the increased interaction and potential rearrangements of the nanoparticles’ surface structures over time due to the unique characteristics of the amphiphilic polymer. The hydrophilic portion of the mPEG, being attracted to water molecules and polar solvents, establishes a strong affinity with the surrounding environment. With an extended hydrophilic chain, more interaction sites are available, influencing the hydrophilic–hydrophobic balance and surface energy of the nanoparticles. The dynamic nature of amphiphilic polymers allows for conformational changes over time, and the longer hydrophilic chains provide increased flexibility, enabling the polymer to adapt to environmental variations during storage.

### 3.3. Characterization of O-I Hybrid TiO_2_ Coating Films

O-I TiO_2_ hybrid sols could be easily applied on various substrates. The curing process also required a low temperature to form a coating on the substrate. Figure 4 illustrates the formation of a homogeneous transparent thin coating film on a substrate from the O-I hybrid titania solution. During the thermal annealing, O-I hybrid TiO_2_ nanoparticles dispersed in organic solvent tended to be close together because of the evaporation of the solvent. At the same time, the post−condensation reaction occurred between the remaining alkoxy groups of nanoparticles within them.

Therefore, the UTi sols could be converted to solid UTi nanoparticles, a process commonly referred to as the sol–gel process [35]. When UTi nanoparticles came close to each other, the amphiphilic polymer that covered the outside of the nanoparticles may have filled the space between the nanoparticles, resulting in the formation of a continuous film on the substrate. Consequently, the AUP not only played a role in the stabilization of the nanoparticles, preventing them from aggregating to macroscale size, but also formed a continuous and homogeneous coating film on the substrate [33]. To determine the composition of the O-I hybrid titania coating film after the thermal annealing process and the inorganic content of the coating films, X-ray photoelectron spectroscopy (XPS) analysis and thermogravimetric analysis were conducted, respectively. Figure 5a,b shows the XPS survey spectra of the UTi7 coating. It can be seen that the dominant elements are O, Ti, C, and N (Figure 5a). Among them, O and Ti had the highest concentrations, which illustrates that the solute element was strongly involved in the formation process of the TiO_2_ coatings. The Ti2p_3/2_ peak of 454.5 eV and the Ti2p_1/2_ peak of 463.4 eV indicate the existence of the TiO_2_ coating, as shown in Figure 5b. The inorganic content measured via TGA in the coatings was 52–53% and was not dependent on the type of amphiphilic polymer structure, indicating that the degree of condensation of the O-I TiO_2_ hybrid was high (Figure 5c).

### 3.4. Electrical Characterization of O-I Hybrid Titania Coatings and Nanocomposite Coatings

To clarify the difference in the dielectric properties between nanocomposites and nanohybrid gate dielectric coatings, our experiment focused on comparing the electrical properties of two distinct formulations: the nanocomposite coating film, composed of TiO_2_ nanoparticles dispersed within the poly(AUP) 1000-m550 polymer matrix; and the O-I hybrid coating film (UTi), characterized by an organic–inorganic hybrid titania solution. The dielectric characteristics of these coating films were evaluated using metal–insulator–metal (MIM)-structured capacitors, where the UTi film was deposited between Al electrodes. Figure 6 shows the optical microscope (OM) images of the nanocomposite films and the O-I nanohybrid films deposited using a spin-coating process after the thermal annealing process. While the TiO_2_/poly(AUP) nanocomposite coating films exhibited numerous small holes, indicating potential challenges in film uniformity, the O-I TiO_2_ hybrid films showed highly uniform and defect-free morphology. Moreover, all the nanocomposite films showed inhomogeneity and unevenness, attributable to the agglomeration of TiO_2_ particles during the dispersion process and the subsequent coating film formation. This can lead to an increase in leakage current when applying the voltage to the coating film and a decrease in its dielectric constant.

Figure 7a,b illustrate the dielectric constants of the O-I TiO_2_ hybrid coating films and nanocomposite TiO_2_/poly(AUP) coating films. The dielectric constant of the O-I TiO_2_ hybrid coating films was significantly higher than that of the TiO_2_/poly(AUP) coating films. Specifically, the dielectric constants of UTi7-m550, UTi7-m750, and UTi7-m2000 were measured at 18.6, 7.79, and 6.24, respectively, at 1 MHz. In contrast, the dielectric constant values for TiO_2_/poly(AUP) nanocomposites with varying TiO_2_ content (5 wt%, 10 wt%, and 15 wt%) were markedly lower, measured at 2.63, 3.57, and 4.22 at 1 MHz, respectively. Figure 7c,d show that the leakage current of the O-I hybrid films was lower than that of the nanocomposite films. The leakage current levels of the O-I hybrid films UTi7-m550, UTi7-m750, and UTi7-m2000 were 2.6 × 10^−7^, 1.97 × 10^−6^, and 3.07 × 10^−5^ at 2 MV·cm^−1^, respectively. In comparison, the leakage current levels of the nanocomposite films of TiO_2_/poly(AUP)—5 wt%, TiO_2_/poly(AUP)—10 wt%, and TiO_2_/poly(AUP)—15 wt% were 1.66 × 10^−6^, 4.03 × 10^−5^, and 1.75 × 10^−3^ at 2 MV·cm^−1^, respectively. These results were expected due to the differences in structure of the two materials. In nanocomposites such as TiO_2_/poly(AUP), the dielectric constant is influenced by the individual dielectric constants of the components and their arrangement within the material. Even TiO_2_, as an inorganic material, generally exhibits a higher dielectric constant compared to organic materials, and the incorporation of TiO_2_ nanoparticles within the polymer matrix can result in a dilution effect, ultimately leading to a reduction in the overall dielectric constant of the composite. Furthermore, the presence of small holes and inhomogeneities in the nanocomposite film might also contribute to a further decrease in the effective dielectric constant [36]. On the other hand, the UTi nanohybrid coating films were designed to have a more integrated and uniform structure, combining organic and inorganic components at the same time, leading to the homogeneity and synergy between the components, potentially resulting in a higher overall dielectric constant for the nanohybrid film compared to the nanocomposite. However, there was a pronounced decrease in all of the nanocomposite coating films and the O-I hybrid TiO_2_ coating film when the frequency was increased to 1 MHz. This could be due to the unreacted hydroxyl groups on the TiO_2_ nanoparticles’ surfaces. At low frequencies, the dipolar groups, such as unreacted hydroxyl groups, have ample time to react to the applied electric field, aligning themselves and thereby contributing to a higher dielectric constant. This alignment significantly boosts the material’s capacity to store electrical energy. However, with an increase in the frequency of the applied electric field, there is insufficient time for the dipoles or charges to reorient in response to the rapidly changing field. Consequently, this leads to a reduction in the dielectric constant as the material becomes less efficient in storing electrical energy [37].

### 3.5. Effect of AUPs on the Dielectric Properties of O-I Hybrid Coatings

Based on the dielectric constant and leakage current profiles in Figure 7a,c, it can be seen that the amphiphilic polymer, particularly the length of its hydrophilic chain, has a strong impact on the dielectric properties of the O-I hybrid coatings. The dielectric constant of the O-I hybrid coatings exhibited an inverse relationship with the length of the hydrophilic chain of the AUP (longer hydrophilic chain resulted in a smaller dielectric constant), while the leakage current value showed a direct proportionality to the length of the hydrophilic chain of the AUP (longer hydrophilic chain correlated with a higher leakage current value). This is because longer hydrophilic chains might lead to more extended and less densely packed structure of nanoparticles within the coatings, thereby limiting the interaction of polarizable groups in response to the applied electric field [30]. Figure 8a illustrates the surface structure of O-I hybrid coatings, characterized by scanning electron microscopy (SEM) analysis. By using the AUP with the shortest hydrophilic chain, the UTi7-m550 coating showed a smooth and denser surface compared to the UTi7-m750 and UTi7-m2000 coatings, allowing for a more efficient alignment of dipoles. This alignment led to a greater degree of polarization and, consequently, a higher dielectric constant. Additionally, the longer hydrophilic chain could exhibit a greater tendency to entangle and form aggregates within the coating, resulting in surface irregularities. The difference in intermolecular interaction between the hydrophilic chain and TiO_2_ nanoparticles also affect the arrangement of polymer molecules, leading to the formation of clusters/domains that contribute to surface roughness. From Figure 8b, it can be seen that the longer hydrophilic chains in the amphiphilic polymer showed higher surface roughness, which provided additional pathways and defects in the coating structure. These pathways can facilitate the movement of charge carriers, promoting a higher leakage current in the film [38].

### 3.6. OTFT Device Performance

To demonstrate the practical application of our prepared material, UTi7-m550 was chosen as the gate dielectric in preparing the OTFT device due to the high dielectric constant with a small leakage current, as measured before. For the fabrication of OTFT arrays, the gate electrodes were prepared with Al deposition, and gate dielectrics were deposited through the solution-processing of UTi7-m550. Following the annealing stage of the sol–gel reaction, the C10-DNTT OSC layers and 50 nm Au source/drain electrodes were applied to finalize the production of the OTFT arrays, as shown in Figure 9a,b. The transport and output current–voltage characteristics of the OTFT with the UTi7-m550 gate dielectric at the saturation region are illustrated in Figure 9c. Due to the high-κ properties of the UTi7-m550 layer, the fabricated OTFT operates in the saturation state at very low operating voltage (V_G_) conditions ranging from 1 to −2 V and a source–drain voltage (V_D_) of −2 V, aiming to achieve transport characteristics under ambient air conditions. The transfer and output curves of this typical p-type OTFT device showed linear and saturation regions with negligible hysteresis, implying reliable transistor operation in the low-voltage operation range. Based on the transport current–voltage properties of UTi7-m550, the electrical parameters of the device were calculated using the following formula:(1)$$I_{D}={\mu FETC_{i}\frac{W}{2L}(V_{G}-V_{th})}^{2}$$
where μFET represents the field-effect mobility, C_i_ is the capacitance per unit area (measured at 50 nF·cm^−2^), and W/L is the ratio between the channel width (1500 μm) and the effective length (150 μm) up to 10. I_D_ and V_th_ denote the drain current and threshold voltage of the device, respectively. The performance of the organic thin-film transistor (OTFT) is summarized in Table 3.

Regarding the output data, the p-type organic thin-film transistor (OTFT) provides considerable insight into the future fabrication of reliable large-area production across key parameters. The average effective field-effect mobility of 6.39 cm^2^·V^−1^·s^−1^ was high, indicating efficient charge carrier transport within the device. This high mobility was reinforced by the on/off ratio of 1.76 × 10^4^, showcasing a distinct separation between conducting and nonconducting states. The negative threshold voltage (V_th_) of −0.95 V aligned with expectations for a p-type transistor, signifying the gate voltage at which conduction initiates. The subthreshold swing (SS) value, at 0.272 V/dec, while not extremely low, was still acceptable, suggesting efficient control over transistor switching and potential suitability for low-power applications. Moreover, as previously mentioned, the minimal hysteresis observed in the transfer curves and the superior operational stability under continuous bias stress conditions, even when subjected to a drive voltage of 2 V during bias stress testing, with a variation in V_th_ of less than 1 V (Figure 9e), indicated a lack of significant trapping phenomena at the interface between the channel and dielectric layer. This was in addition to the dipole disorder effects discussed earlier.

## 4. Conclusions

In summary, we successfully prepared low-temperature (<100 °C) solution-processed organic–inorganic TiO_2_ hybrid nanoparticles. The O-I TiO_2_ hybrid nanoparticles could be dispersed in an organic solvent and remain stable for more than 6 months because of the presence of an amphiphilic urethane nonionomer precursor (AUP). The AUP not only provided a smooth and homogeneous coating layer on the substrate but also had strong effects on the dielectric properties of the coating. The O-I TiO_2_ hybrid coating film with the AUP with the shortest hydrophilic chain (UTi7-m550) showed the smallest and most uniform nanoparticles compared to other coating films. The UTi7-m550 exhibited the best electrical properties compared to the TiO_2_/poly(AUP) nanocomposites and the remaining two O-I TiO_2_ hybrid materials. The dielectric constant of the UTi7-m550 dielectric layer was found to be 18.6 at 1 MHz, and the dielectric layer exhibited a low leakage current density of 2.6 × 10^−7^ A/cm^2^ at 2 MV·cm^−1^. The OTFT device that used UTi7-m550 as the dielectric layer demonstrated a high average field-effect mobility of 6.39 cm^2^·V^−1^·s^−1^ with negligible hysteresis observed in the transfer curves and superior operational stability under continuous bias stress conditions. Consequently, we believe that our research on the preparation of colloidally stable O-I TiO_2_ hybrid nanoparticles provides valuable insights for designing high-κ dielectric materials to enhance the operational stability and electrical performance of OTFTs.

## Figures and Tables

**Figure 1 nanomaterials-14-00488-f001:**
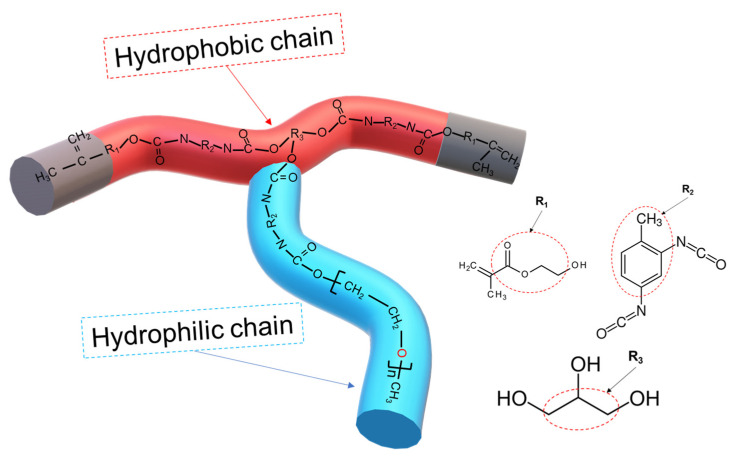
Schematic presentation of the amphiphilic urethane polymer (AUP).

**Figure 2 nanomaterials-14-00488-f002:**
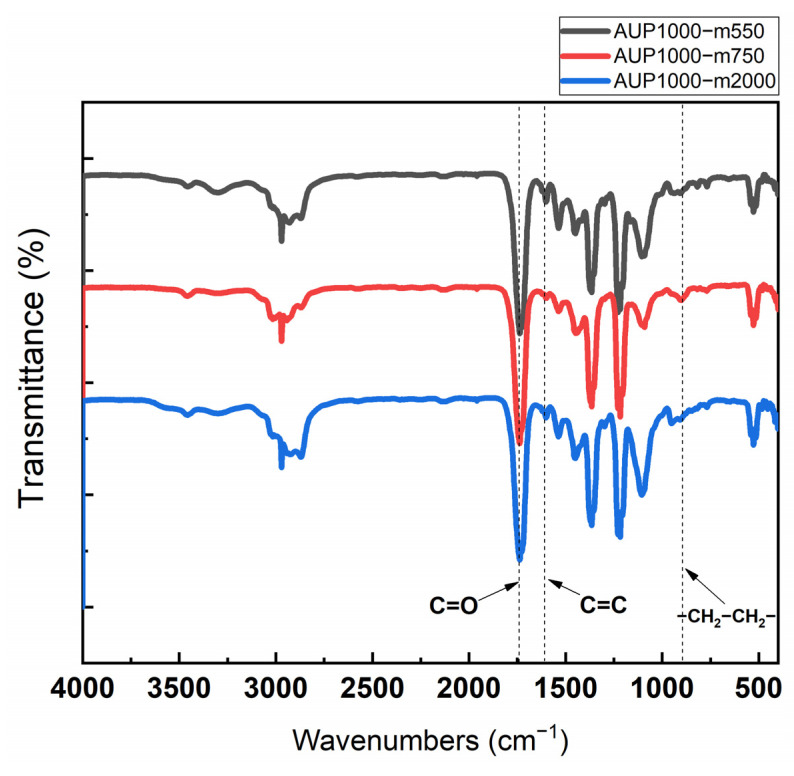
FT-IR spectra of the three AUPs.

**Figure 3 nanomaterials-14-00488-f003:**
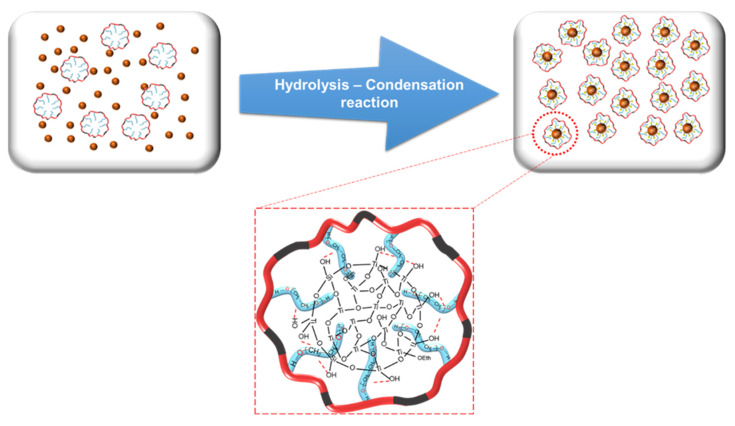
Preparation of O-I TiO_2_ hybrid nanoparticles in the presence of AUP.

**Figure 4 nanomaterials-14-00488-f004:**
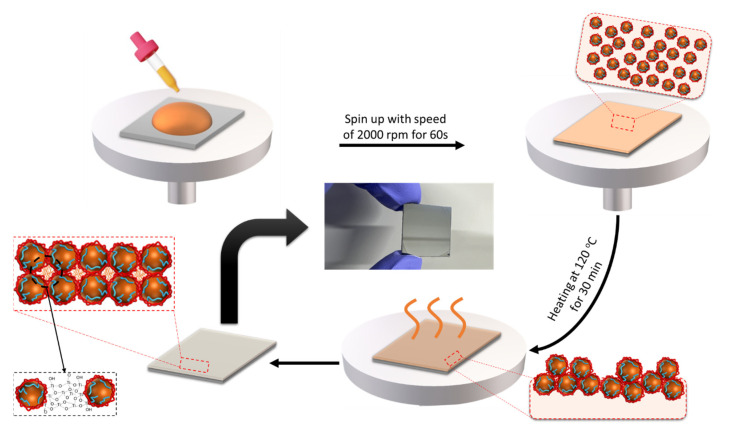
Formation mechanism of O-I hybrid coating films.

**Figure 5 nanomaterials-14-00488-f005:**
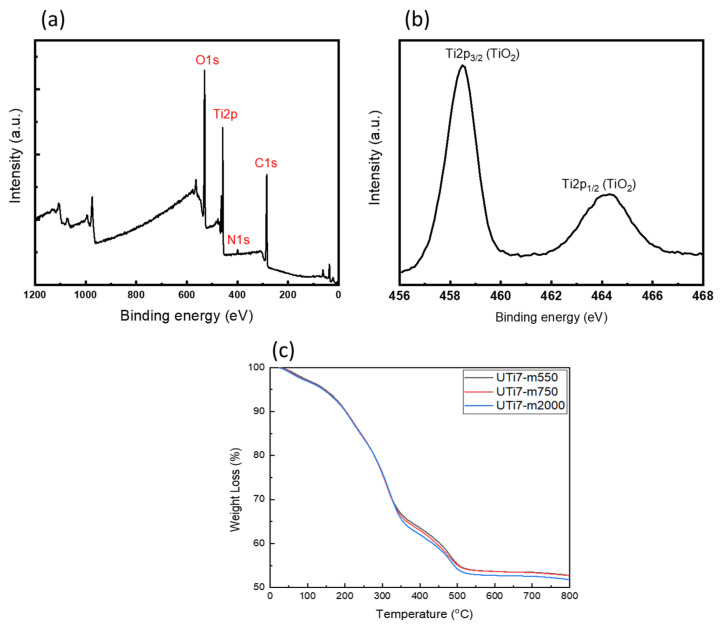
XPS data spectra of (**a**) typical survey of O-I TiO_2_ hybrid coating and (**b**) Ti2p; (**c**) TGA data.

**Figure 6 nanomaterials-14-00488-f006:**
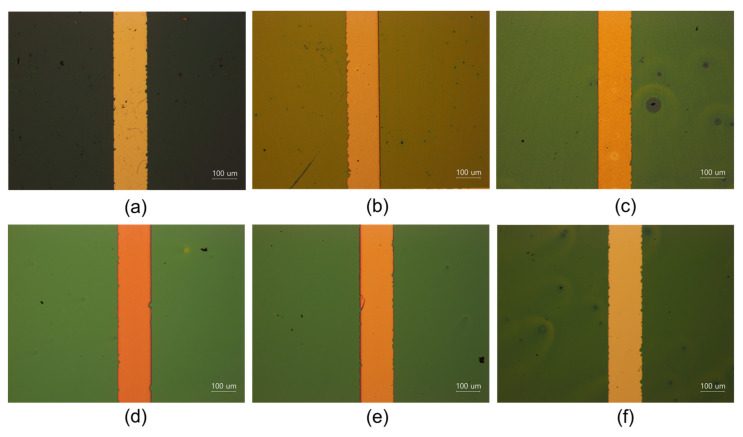
Optical microscope images. (**a**) TiO_2_/poly(AUP)—5 wt% coating; (**b**) TiO_2_/poly(AUP)—10 wt% coating; (**c**) TiO_2_/poly(AUP)—15 wt% coating; (**d**) UTi7-m550 coating; (**e**) UTi7-m750 coating; (**f**) UTi7-m2000 coating.

**Figure 7 nanomaterials-14-00488-f007:**
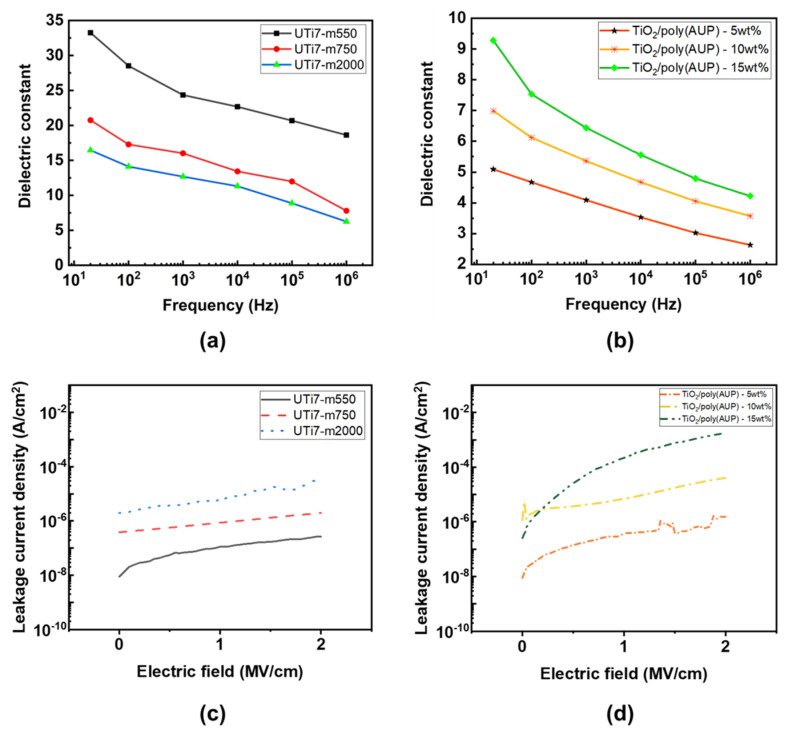
Dielectric constant of (**a**) O-I hybrid coating and (**b**) nanocomposite coating according to frequency; (**c**,**d**) leakage current density properties of O-I hybrid coating and nanocomposite coating, respectively.

**Figure 8 nanomaterials-14-00488-f008:**
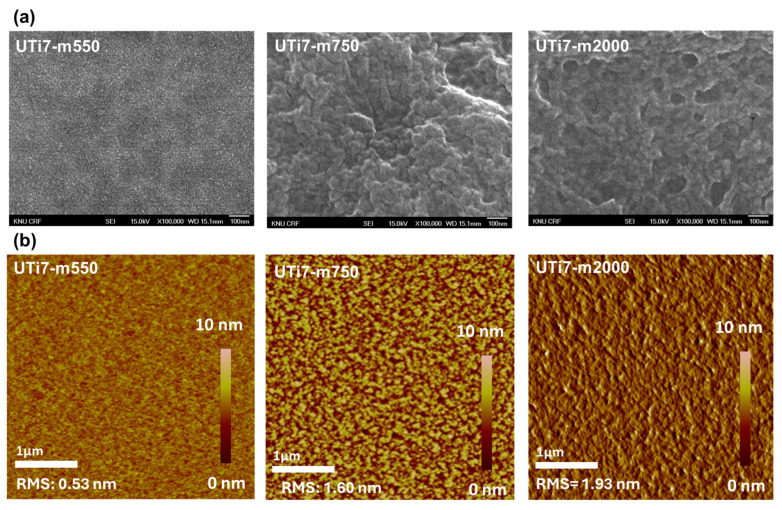
(**a**) SEM images and (**b**) AFM topographic images of O-I hybrid coatings.

**Figure 9 nanomaterials-14-00488-f009:**
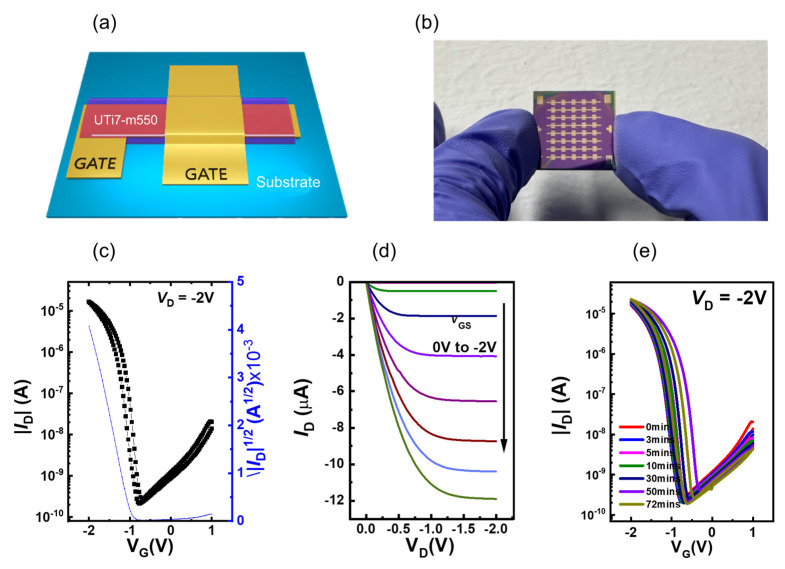
(**a**) Device schematic with UTi7-m550 as gate dielectric; (**b**) photographs of the large-scale OTFT arrays on Si wafer when using the spin-coating method; (**c**) electrical characteristics of C10-DNTT OTFT devices; (**d**) dielectrics’ transfer curve; (**e**) transfer curve with bias stress test.

**Table 1 nanomaterials-14-00488-t001:** Molecular weight and nanoparticle size of AUPs measured with GPC and DLS characterization.

AUPs	Mw (g/mol)	PDI	Particle Size of AUPs Dispersed in Ethanol (nm)
AUP 1000-m2000	38,765	2.1	33.60 ± 0.2
AUP 1000-m750	9060	1.33	9.21 ± 0.07
AUP 1000-m550	8996	1.32	4.74 ± 0.01

**Table 2 nanomaterials-14-00488-t002:** Size of TiO_2_ sols when prepared and 6 months after preparation.

O-I TiO_2_ Hybrid Sols	Size of TiO_2_ Nanoparticles after Preparation (nm)	Size of TiO_2_ Nanoparticles 6 Months after Preparation (nm)
UTi7-m2000	9.70 ± 0.11	10.3 ± 0.44
UTi7-m750	4.36 ± 0.04	4.44 ± 0.10
UTi7-m550	4.35 ± 0.02	4.37 ± 0.01

**Table 3 nanomaterials-14-00488-t003:** OTFT device output data.

Semiconductor		Circumstance	C_i_ (nF·cm^−2^)	μ_FET_ (cm^2^·V^−1^·s^−1^)	V_th_ (V)	I_on_/I_off_	SS (V/dec)
C_10_-DNTT	1-2	SiO_2_—100 nm	5.01 × 10^−7^	6.39	−0.95	1.76 × 10^4^	0.272

## Data Availability

Data are contained within the article.

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
