# Peer review of "Preparation of Low-Temperature Solution-Processed High-κ Gate Dielectrics Using Organic–Inorganic TiO2 Hybrid Nanoparticles"

_nanomaterials, 2024, doi:10.3390/nano14060488_

Round 1
Reviewer 1 Report
Comments and Suggestions for Authors
Review for manuscript: Preparation of low-temperature solution-processed high-κ gate dielectrics using organic–inorganic TiO2 hybrid nanoparticles
Manuscript ID : nanomaterials-2896054
The manuscript describes an interesting subject from the field of organic-inorganic hybrids, a novel class of materials with promising applications in different areas. In the last decade, the interest for such hybrid compounds increased significantly and it is still growing. The method used for their synthesis is also very important, while the sol-gel process takes place generally at low temperatures, by using solvents with lower impact on the environment. Therefore, the sol-gel method is of great interest nowadays and is a green chemistry synthesis.
The Abstract is well presented and the goal of the work is clear for the readers. It is important for the authors to decrease the percentage resulting from the ithenticate report (32%).
The Introduction could and should be improved, by citing more bibliographic sources in the fields of sol-gel synthesis of organic-inorganic hybrids, as the work of Gheonea et al.: New hybrid materials synthesized by sol-method, Advances in Materials Science and Engineering, 2017 https://doi.org/10.1155/2017/4537039. This work describes the use of sol-gel synthesis to obtain organic-inorganic hybrids containing phosphorus and zirconium. The use of zirconium is another option, sometimes used even to replace titanium. Both, Ti and Zr have strong mechanical properties and good resistance, and also they have similar chemical behavior. Therefore such work should be included in the Introduction. Nevertheless, the class of organic-inorganic hybrids phosphorus-containing is of great importance, and should be mentioned in the Introduction.
The methods and the used reagents and precursors are well described. The Results and Discussion section could be improved. Here the authors can also make a discussion related to the comparison of using Ti, and why choosing Ti, in comparison with other possibilities, as Zr, Sn, and so on. There are examples in the literature related to the fields of organic-inorganic hybrids containing also other chemical elements.
The GPC chromatographic technique is mentioned and a concentration of 3 mg/ml was used for it. But immediately then, DLS is mentioned for dispersed nanoparticles in THF. For GPC also THF was used. So, there is a solution or a dispersion? This could be confusing for the reader and in general on chromatographic separations, the solutions are involved. Please specify how the solution is obtained for GPC, if it is indeed a solution, and also how the dispersed nanoparticles are obtained for DLS. These two are different systems (in the work described here) or there is the same sample? Generally, the discussion is about the colloidal dispersion. Therefore, is there any solution or not? Because a dispersion and therefore a colloid, is not a solution. All of these should be explained and should be clear. In general, the hybrids are not soluble or they have some solubility, but rather low.
Then, the authors show optical microscopy images, but also SEM and AFM results. If you have the electronic microscope images, with better resolution, what is the point of showing also optical microscopy results?
Morever, the SEM images from Fig. 7a were of good quality, except Uti7-m550. Since the materials are dielectric, therefore they have no conductivity at all. As a consequence, I would expect all electron microscopy images to be very bad and weak from quality and resolution points of view. Because, the most important requirement for a material, to be analyzed by SEM or TEM with good results and resolution, is to have high conductivity. For the materials with low conductivity and for dielectric materials with no conductivity, a special procedure of sputtering is necessary (covering the dielectric material with a thin conductive film). Such a procedure is not described in the manuscript. Therefore, please specify how this good quality of the SEM images was obtained. Are those materials really dielectric? It would be very interesting to analyze the organic-inorganic hybrid materials also by TEM, and by EDX too. Did you consider those techniques and do you plan to use them in the future?
The Conclusions are sustained by the presented results. The quality of the English is good and needs only minor editing. Overall, the manuscript needs several modifications and improvements, adding more citations relevant to the fields of organic-inorganic hybrids synthesized by using the sol-gel method, more discussions, and also adding more details about the analytical methods used and involved in this study (including the sample preparation for GPC and DLS, if there is a solution or not, and how low is the conductivity and how it was possible to obtain such good SEM images for a dielectric material with no conductivity without sputtering them and so on).
After performing the minor revision according to the above-mentioned suggestions and recommendations, the manuscript could be considered for acceptance to be published in Nanomaterials.

Review for manuscript: Preparation of low-temperature solution-processed high-κ gate dielectrics using organic–inorganic TiO2 hybrid nanoparticles
Manuscript ID : nanomaterials-2896054
The manuscript describes an interesting subject from the field of organic-inorganic hybrids, a novel class of materials with promising applications in different areas. In the last decade, the interest for such hybrid compounds increased significantly and it is still growing. The method used for their synthesis is also very important, while the sol-gel process takes place generally at low temperatures, by using solvents with lower impact on the environment. Therefore, the sol-gel method is of great interest nowadays and is a green chemistry synthesis.
The Abstract is well presented and the goal of the work is clear for the readers. It is important for the authors to decrease the percentage resulting from the ithenticate report (32%).
The Introduction could and should be improved, by citing more bibliographic sources in the fields of sol-gel synthesis of organic-inorganic hybrids, as the work of Gheonea et al.: New hybrid materials synthesized by sol-method, Advances in Materials Science and Engineering, 2017 https://doi.org/10.1155/2017/4537039. This work describes the use of sol-gel synthesis to obtain organic-inorganic hybrids containing phosphorus and zirconium. The use of zirconium is another option, sometimes used even to replace titanium. Both, Ti and Zr have strong mechanical properties and good resistance, and also they have similar chemical behavior. Therefore such work should be included in the Introduction. Nevertheless, the class of organic-inorganic hybrids phosphorus-containing is of great importance, and should be mentioned in the Introduction.
The methods and the used reagents and precursors are well described. The Results and Discussion section could be improved. Here the authors can also make a discussion related to the comparison of using Ti, and why choosing Ti, in comparison with other possibilities, as Zr, Sn, and so on. There are examples in the literature related to the fields of organic-inorganic hybrids containing also other chemical elements.
The GPC chromatographic technique is mentioned and a concentration of 3 mg/ml was used for it. But immediately then, DLS is mentioned for dispersed nanoparticles in THF. For GPC also THF was used. So, there is a solution or a dispersion? This could be confusing for the reader and in general on chromatographic separations, the solutions are involved. Please specify how the solution is obtained for GPC, if it is indeed a solution, and also how the dispersed nanoparticles are obtained for DLS. These two are different systems (in the work described here) or there is the same sample? Generally, the discussion is about the colloidal dispersion. Therefore, is there any solution or not? Because a dispersion and therefore a colloid, is not a solution. All of these should be explained and should be clear. In general, the hybrids are not soluble or they have some solubility, but rather low.
Then, the authors show optical microscopy images, but also SEM and AFM results. If you have the electronic microscope images, with better resolution, what is the point of showing also optical microscopy results?
Morever, the SEM images from Fig. 7a were of good quality, except Uti7-m550. Since the materials are dielectric, therefore they have no conductivity at all. As a consequence, I would expect all electron microscopy images to be very bad and weak from quality and resolution points of view. Because, the most important requirement for a material, to be analyzed by SEM or TEM with good results and resolution, is to have high conductivity. For the materials with low conductivity and for dielectric materials with no conductivity, a special procedure of sputtering is necessary (covering the dielectric material with a thin conductive film). Such a procedure is not described in the manuscript. Therefore, please specify how this good quality of the SEM images was obtained. Are those materials really dielectric? It would be very interesting to analyze the organic-inorganic hybrid materials also by TEM, and by EDX too. Did you consider those techniques and do you plan to use them in the future?
The Conclusions are sustained by the presented results. The quality of the English is good and needs only minor editing. Overall, the manuscript needs several modifications and improvements, adding more citations relevant to the fields of organic-inorganic hybrids synthesized by using the sol-gel method, more discussions, and also adding more details about the analytical methods used and involved in this study (including the sample preparation for GPC and DLS, if there is a solution or not, and how low is the conductivity and how it was possible to obtain such good SEM images for a dielectric material with no conductivity without sputtering them and so on).
After performing the minor revision according to the above-mentioned suggestions and recommendations, the manuscript could be considered for acceptance to be published in Nanomaterials.
Author Response
Thank you very much for taking the time to review this manuscript. We are grateful to the reviewer for acknowledging the strengths of our research and offering valuable feedback. We have made the required modifications to our manuscript based on the comments and useful suggestions provided by you. Please find the detailed responses below and the corrections highlighted in the re-submitted files.

Reviewer 2 Report
Comments and Suggestions for Authors
The article “Preparation of low-temperature solution-processed high-κ gate 2 dielectrics using organic–inorganic TiO2 hybrid nanoparticles” is dealing with the preparation of colloidally stable 0-I TiO2 hybrid nanosols in the presence of an amphiphilic urethane precursor. Fabrication of MIM and OTFT devices to assess the insulating properties of the I-O TiO2 dielectric layers (coating) and the operational stability and electrical performance of OTFTs is performed. Characterization of 0-I hybrid TiO2 nanosols and coating films including electrical characterization was also carried out.
However, there are issues which require more explanation, additional data, including re-interpretation of data and reformulation of statements. Numbering of Figures should corrected.
If authors are willing to consider these comments and questions and perform an in-depth revision, the revised paper can be reconsidered for publication.
A detailed list of comments and questions is addressed to the authors.
2. Materials and Methods
Line 144: Conditions of polymerization of AUP in MEK should be more detailed. Please comment.
Line 150: What is the pH of to complete the hydrolysis-condensation reaction? Please comment.
Line 157: Concentration, purity and supplier of alkaline H2O2 solution should be included in section 2.1. Please comment.
Line 160: “The resulting precipitate was thoroughly ionized”. This formulation is not clear to me. Please comment.
Line 162: “70 mL” instead of “70 ml”
Line 165: CY is not defined. Please comment.
Lines 175 and 180: Supplier and purity of acetone, isopropanol and 2-methoxyalcohol should be included in section 2.1. Please comment.
Line 187: “(2,9-di-decyl-dinaphtho[2,3-b:2',3'-f]-thieno-[3,2-b]-thiophene” instead of “(2,9-di-decyll-dinaphtho[2,3-b:2',3'-f]-thieno-[3,2-b]-thiophene”
Lines 196-197: More information on experimental conditions of FTIR analysis is required: ATR? Number of scans? Resolution? Detector? Please comment.
Lines 198-200: Which calibration standards were used for GPC analysis? Calibration range? Please comment.
Line 206: Flow of N2 atmosphere in TGA? Please comment.
Line 219: Fig.1 R is not defined in Fig.1. “R-O-H –On” as such is not possible unless it should be “R-O-H---On” (H bridging). Please comment.
3. Results and Discussion
Line 234: Abbreviation PDI (polydispersity index) should be explained when first introduced. Please comment.
Line 246: In Table 1: Number of digits in standard deviations should be the same as in the mean. E.g. “33.6±0.2” instead of “33.6±0.21”. Please comment.
Line 254: “titanol” instead of “tilanol”
Line 255: “r value” is not explained. Please comment.
Line 256: “Ti-O-Ti crosslinks by the dealcoholation reaction of the dehydration reaction” is not clear. There is first a dealcoholation reaction followed by the dehydration reaction of the formed Ti-OH groups. Please comment.
Lines 276-277: Considering the standard deviations (Table 2) there is no statistical difference in the particle size after 6 months, especially for UTi7-m750 and UTi7-m550. Please comment.
Line 287: Table 1 should be Table 2. Please comment.
Line 290: Considering the standard deviations (Table 2) there is no statistical difference between both particle sizes. Please comment.
Line 302: In Table 1: Number of digits in standard deviation should be the same as in the mean. E.g. “4.44±0.10” instead of “4.44±0.096”. In addition, number of digits should be limited to three significant digits. Please comment.
Lines 320-331: When comparing peak intensities in FTIR, absorbance should be used and not % transmission (Beer’s law). Vertical label is missing in Fig. 5 (a). Please comment. The peak at 1624 cm-1 of TiOH assigned cannot be differentiated from O-H of ethanol or of moisture (bending vibration of O-H). I do not see any increase of the peak at 483 cm-1. Please comment. Ti-O-Ti peaks are broad and are within the 800-400 cm-1 range. Please comment.
Line 333: “thermogravimetric analysis” instead of “thermal gravity analysis”
Line 335: Fig.5b does not contain any information that “It can be seen that the dominant elements are O, Ti, C, and N". Such information should be included. Please comment.
Line 337: The Ti2p3/2 peak at 454.27 eV does not fit with the XPS spectrum in Fig5b and should be 458.2 eV. Please comment.
Line 338: The peak at 463.b is not corresponding with Ti2p3/2 of TiO2 but with Ti2p1/2 of TiO2. Please comment.
Lines 338-340: Which type of TiO2 is obtained? (Anatase or Rutile?). Please comment.
Line 363: A scale should be included in Fig.6. Please comment.
Lines 404 and 408: Figure 6 should be Figure 7.
Line 428: Figure 7 should be Figure 8.
Lines 432 and 444: Figure 8 should be Figure 9.
Line 457: The number of significant digits for µFET should be limited. Please comment.
Comments on the Quality of English Language
Minor editing of English language is required (see detailed list of comments)
Author Response
We extend our sincere gratitude for investing time in reviewing our manuscript. We appreciate the detailed comments, which not only recognized the strengths of our research but also provided valuable feedback. Your insights have been instrumental in shaping the improvements made to our manuscript. Please find below the detailed responses and observe the corresponding corrections highlighted in the re-submitted file.

Reviewer 3 Report
Comments and Suggestions for Authors
1) What is the structure of the nanocomposite? Id the PU chemically bound or just adsorbed on the TiO2?
2) In section 2.5.1, the elaborate substrate cleaning will not remove metallic contaminants; it just remove organics from the substrate's surface.
3) line 160: what does thoroughly ionized mean in this context?
4) This paper has some made up terms, e.g., line 254, calling Ti-OH tilanol is very confusion.
5) Line 287: Table 1 should be Table 2
6) Line 302, table 2 caption: should it be 4 or 6 months. Reconcile with text.
7) Line 315, please provide reference for the "sol-gel transition" statement
8) If the films in the MIM structure have pinholes, how can you trust the electrical data?

line 358: delete the word "likely". It makes the sentence too speculative.
Author Response
Thank you sincerely for dedicating your time to review our manuscript. We highly value the comprehensive examination and constructive feedback you provided. Each of your comments has been thoughtfully considered, and we have implemented necessary modifications accordingly. Please find the detailed responses below, and the corresponding corrections have been highlighted in the re-submitted file.
*** Please see the attachment***

Round 2
Reviewer 2 Report
Comments and Suggestions for Authors
I have read the revised article “Preparation of low-temperature solution-processed high-κ gate 2 dielectrics using organic–inorganic TiO2 hybrid nanoparticles”, including the answers to the reviewers.
Authors have addressed the comments and questions of the reviewers and have implemented changes and additional data and information. I am convinced that the overall quality of the article has improved considerably.
Therefore I support publication of this manuscript after minor corrections.
Minor comments
Line 141: “used in washing step. Finally, for device preparation” instead of “used in washing step Finally, for device preparation”
Line 219: “R-O-H –On” as such is not possible. In my opinion it should be
R-O-[CH2-CH2-O]n-CH3
Author Response
We extend our sincere gratitude for investing time in reviewing our manuscript. Thank you for your positive feedback and support. We appreciate your careful review and have made the suggested corrections. Please find below the detailed responses and observe the corresponding corrections highlighted in the re-submitted files.
